# A Label-Free Colorimetric Aptasensor for Flavokavain B Detection

**DOI:** 10.3390/s25020569

**Published:** 2025-01-19

**Authors:** Sisi Ke, Ningrui Wang, Xingyu Chen, Jiangwei Tian, Jiwei Li, Boyang Yu

**Affiliations:** 1Jiangsu Key Laboratory of TCM Evaluation and Translational Research, School of Traditional Chinese Pharmacy, China Pharmaceutical University, Nanjing 211198, China; wy893406895@163.com (S.K.); chenxingyugs@163.com (X.C.); boyangyu59@163.com (B.Y.); 2School of Laboratory Medicine, Nanchang Medical College, Nanchang 330052, China; aurora_0711@163.com

**Keywords:** flavokavain B, aptamer, aptasensor, gold nanoparticle, colorimetric assay

## Abstract

Flavokavain B (FKB), a hepatotoxic chalcone from *Piper methysticum* (kava), has raised safety concerns due to its role in disrupting redox homeostasis and inducing apoptosis in hepatocytes. Conventional chromatographic methods for FKB detection, while sensitive, are costly and impractical for field applications. In this work, DNA aptamers were selected using the library-immobilized method and high-throughput sequencing. Three families of aptamers were obtained, and the best one named FKB-S showed a dissociation constant (K_D_) of 280 nM using microscale thermophoresis. To demonstrate its practical utility, a rapid and label-free colorimetric aptasensor was developed based on aptamer-induced gold nanoparticle aggregation. This assay achieved a detection limit of 150 nM (43.46 ng/mL) and provided results within 10 min. Compared to traditional chromatographic methods, the aptasensor offers a simple, cost-effective, and equipment-free approach for on-site FKB detection, making it a promising tool for the quality control and safety monitoring of kava-based products in diverse environments.

## 1. Introduction

Flavokavain B (FKB) is a chalcone compound extracted from the rhizomes of kava (*Piper methysticum*), which is a plant recognized for its potential hepatotoxicity. Kava, a member of the Piperaceae family, is a perennial shrub native to South Pacific islands such as Vanuatu, Fiji, and Tonga [1,2]. For the indigenous peoples of the Pacific Rim and the Hawaiian Islands, kava holds profound cultural and medicinal significance. Traditionally, its roots or rhizomes are used to prepare beverages that relax muscles, reduce stress, restore energy, and relieve pain. These beverages are integral to religious ceremonies, celebrations, and rites of passage [3,4]. Beyond its role as a daily beverage and recreational drink, kava has been developed into pharmaceutical formulations due to its unique pharmacological properties. It is commonly used to treat anxiety, menopausal symptoms, insomnia, and seizures [5,6]. Compared to conventional anxiolytics, kava-based preparations offer significant advantages, including the absence of addictive potential, hallucinogenic effects, and other common side effects. Consequently, since the 1990s, countries such as Germany, the United States, and the United Kingdom have approved kava for use as an anxiolytic, leading to its widespread availability and use in the market.

With the growing use of kava products, severe hepatotoxic side effects have been increasingly reported in Europe and the United States. These adverse effects include liver failure, acute severe hepatitis, pancreatic necrosis, disruption of hepatic lobular structure, and hepatocyte apoptosis associated with elevated bilirubin levels. In some cases, these conditions have resulted in death or necessitating liver transplantation [7]. Research has identified chalcone compounds from the rhizomes of the kava plant as the primary contributors to these hepatotoxic events, particularly flavokavain A (FKA), FKB, and flavokavain C (FKC). Among them, FKB exhibits the highest hepatotoxicity [8]. As a potent glutathione (GSH)-sensitive hepatotoxin, FKB depletes intracellular reduced GSH during metabolism, converting it into oxidized glutathione (GSSH). This depletion severely disrupts the redox homeostasis of hepatocytes and further triggers apoptosis through the activation of the IKK/NF-κB and MAPK signaling pathways [9].

Given the rising incidence of kava-related hepatotoxicity events, it is urgent to incorporate FKB alongside kavalactones into the new quality control standards for kava [10,11]. Currently, the analysis and detection of kava-based formulations and products primarily rely on liquid chromatography techniques and their coupled methods [12,13]. While these instrumental approaches offer excellent sensitivity and accuracy, they face challenges in terms of practicality and cost-effectiveness in real-world applications. Therefore, the development of a novel, efficient, and cost-effective method for FKB detection has become an urgent research priority. More recently, aptamer-based detection has gained popularity. Aptamers have emerged as highly promising molecular recognition elements for detecting FKB. These nucleic acid-based bioreceptors are selected from randomized oligonucleotide libraries through a systematic in vitro method known as SELEX (Systematic Evolution of Ligands by Exponential Enrichment), enabling them to bind specifically to target molecules [14,15,16]. In addition to their high affinity and specificity, which are comparable to those of antibodies, aptamers exhibit exceptional chemical stability, low production costs, excellent batch-to-batch consistency, and the ability to remain stable in powdered form, facilitating easy storage and transport [16,17]. Thus, developing a novel, efficient, and cost-effective method for detecting FKB has become a top research priority.

Among the various aptamer-based sensing platforms developed, colorimetric assays stand out for on-site applications due to their ease of use and visual readout, requiring no specialized equipment [18,19,20]. In particular, gold nanoparticles (AuNPs) are frequently integrated with aptamers as highly sensitive visual reporters, facilitating the direct detection of small molecules [21,22]. In this study, we first isolated an FKB-binding DNA aptamer utilizing a combination of library-immobilized SELEX and high-throughput sequencing (Figure 1). The truncated aptamer, termed FKB-S, exhibited nanomolar K_D_ (280 nM) and high specificity for FKB with minimal cross-reactivity to potential interferents. We then demonstrated the aptamer’s on-site applicability by developing a simple aptamer/gold nanoparticle aggregation-based colorimetric assay for FKB determination. This assay is straightforward, involving just a swift mixing step in a microplate, and provides results within 10 min with a detection threshold as low as 150 nM (43.46 ng/mL). Unlike previously reported chromatography, this assay offers a rapid, accurate, and equipment-free approach for onsite FKB detection. Its simplicity, swiftness, and cost-effectiveness render it exceptionally suitable for field applications, particularly in environments with limited access to sophisticated equipment. The method holds immense potential for widespread adoption in monitoring kava products and assessing their safety.

## 2. Materials and Methods

### 2.1. Chemicals and Reagents

The oligonucleotides (Appendix A) and the UNIQ-10 Spin Column DNA Gel Extraction Kit were obtained from Sangon Biotech (Shanghai, China). Flavokavain B, myricetin, quercetin, aloe emodin, isoliquiritigenin, flavokavain A, flavokavain C, hyperoside, puerarin, hesperidin, astilbin, catechin, ophiopogonone B, acrylamide, bis-acrylamide, formamide, and urea were purchased from the Aladdin Industrial Corporation (Shanghai, China). Ammonium persulfate and tetramethylethylenediamine were supplied by Solarbio (Beijing Solarbio Science & Technology Co., Ltd., Beijing, China). SYBR Gold nucleic acid stain (10,000 × in DMSO) was procured from Invitrogen (Grand Island, NY, USA). Streptavidin-coated agarose resin was obtained from Thermo Fisher Scientific. Micro Bio-Spin chromatography columns (500 μL) were sourced from Bio-Rad, and 2 × GoTaq Hot Start Colorless Master Mix was acquired from Promega. Spin filters with a 3 kDa molecular weight cut-off were purchased from Millipore, and Monolith NT™ Standard Treated Capillaries were supplied by NanoTemper Technologies. Gold acid chloride trihydrate (HAuCl_4_·3H_2_O) was obtained from Aladdin Bio-Chem Technology Co., Ltd. (Shanghai, China).

### 2.2. Apparatus

The concentrations of all nucleic acids were determined using a NanoDrop spectrophotometer (Thermo Fisher Scientific, Waltham, USA). PCR amplifications were conducted on a T100 Thermal Cycler (Bio-Rad, Hercules, CA, USA). Microscale thermophoresis (MST) measurements were performed with a Monolith NT.115 instrument (NanoTemper Technologies GmbH, Munich, Germany). Absorbance readings were obtained using an Infinite M200 Pro TECAN plate reader (Tecan, Mannedorf, Switzerland).

### 2.3. SELEX Procedure

SELEX was carried out as previously described [23,24] with detailed screening conditions outlined in Appendix A. Briefly, the 73 nt stem-loop structured library or the enriched pool from the previous round was mixed with a biotinylated complementary strand (cDNA-bio) in selection buffer (10 mM Tris-HCl, 0.5 mM MgCl_2_, 20 mM NaCl, pH 7.4). The mixture was heated to 95 °C for 5 min and then gradually cooled to room temperature over 30 min. The annealed complex was loaded into a microgravity column containing streptavidin-coated agarose resin for library immobilization. Unbound library and complementary strands were removed by washing the column with selection buffer at least 5 times. Aptamers bound to the target were eluted with selection buffer containing FKB at varying concentrations: 500 μM (Rounds 1–2), 250 μM (Rounds 3–4), 125 μM (Rounds 5–14), 50 μM (Round 15), and 5 μM (Rounds 16–27). The collected aptamers were concentrated using a 3K molecular weight cut-off spin filter and amplified by PCR. The PCR mixture contained 1 μM forward primer (FP), 1 μM biotinylated reverse primer (RP-bio), and 1 mL of GoTaq Hot Start Master Mix, and it was run on a BioRad T100 thermal cycler. The resulting double-stranded PCR product was immobilized on a streptavidin-coated agarose resin column, which was followed by incubation with 0.2 M NaOH for 10 min to denature the strands. The eluted single-stranded DNA was neutralized with 0.2 M HCl and concentrated using a 3K molecular weight cut-off spin filter. The concentration of the enriched pool was measured with a NanoDrop 2000, and the pool was used for the next SELEX round.

Counter-SELEX was conducted using a panel of compounds. Specifically, from Rounds 6–19, we used the following counter-SELEX strategy: a mixture of 100 µM myricetin, quercetin, aloe emodin, and isoliquiritigenin for Rounds 6–9, a mixture of 100 µM myricetin, quercetin, aloe emodin, isoliquiritigenin, FKA, and FKC for Rounds 10–12, a mixture of 100 µM myricetin, quercetin, aloe emodin, isoliquiritigenin, FKA, FKC, hyperoside, puerarin, hesperidin, astilbin, catechin, and ophiopogonone B for Rounds 13–19, and a mixture of 100 µM FKA and FKC for Round 20–27. The library-immobilized column was first washed three times with 100 μM of each counter-SELEX target in the selection buffer, which was followed by 20 washes with selection buffer alone before eluting the target-bound aptamers.

### 2.4. High-Throughput Sequencing and Bioinformatics Analysis

The round 19 and 27 pools from the SELEX experiment were prepared for sequencing and sent to Sangon Biotech Co., Ltd. for sequencing on an Illumina high-throughput sequencing platform (HiSeq/MiSeq). The raw sequencing reads were processed using Cutadapt for trimming and quality assessed with Prinseq-lite. Sequence population analysis and clustering were performed using AptaSUITE software [25]. The seed sequences from the selected clusters were chosen for secondary structure prediction using a free energy minimization algorithm available in the IDT Oligo Analyzer (1 June 2024. https://www.idtdna.com/calc/analyzer). Subsequently, AlphaFold 3 was used to predict the 3D structure. The 3D conformer of FKB was obtained from the PubChem chemical compound database (PubChem CID: 5356121). Docking simulations were conducted using the HDOCK server, generating ten conformations with the optimal docking results visualized in PyMOL [26].

### 2.5. Gel Elution Assay

A reported gel elution assay was carried out to monitor the target-binding affinity and the specificity of the enriched pools (5th and 19th rounds) or the specific sequenced strands (FKB-1, FKB-2 and FKB-3) [27]. The procedure was as follows: the enriched library or the specific sequenced strands was mixed with cDNA-bio in selection buffer, annealed at 95 °C for 10 min, and cooled to room temperature for over 30 min. The annealed complex was loaded into a microgravity column containing streptavidin-coated agarose resin for library immobilization. The beads were aliquoted into seven PCR tubes. After discarding the supernatant, varying concentrations of FKB or interferent (0–1000 μM) were added, which was followed by 1 h incubation and 15 min standing. Next, supernatant was collected. To release remaining DNA from the beads, denaturing solution (98% formamide, 10 mM EDTA) was added and heated at 90 °C for 10 min, which was followed by 10 min standing to collect the solution. We analyzed the target-eluted aptamer solution and formamide-treated library solution via 15% denaturing polyacrylamide gel electrophoresis (PAGE) and determined the elution percentage using the following equation:$$\theta =\frac{V1\times Cs}{V2\times Cs+V3\times Cb}\times 100\%$$
where θ represents the fraction of target-eluted strands, Cs is the concentration of target-eluted strands in the supernatant, Cb is the concentration of strands in the formamide solution, V1 is the initial solution volume before supernatant collection, V2 is the volume of the collected supernatant containing target-eluted strands, and V3 is the volume after adding the formamide solution. To determine the dissociation constant (K_D_), the data were fitted to a 1:1 Langmuir binding model using OriginPro 2021 software.

### 2.6. Microscale Thermophoresis Assay

Affinity measurements were conducted using Microscale Thermophoresis on a Monolith NT.115 instrument [28]. The aptamer, labeled at the 5′ end with a Cy5 dye (Appendix A, FKB-1-5′FAM or FKB-S-5′FAM), was dissolved in MST buffer (10 mM Tris-HCl, 20 mM NaCl, 0.5 mM MgCl_2_, 5% DMSO, 0.05% Tween-20, pH 7.4). The solution was heated at 95 °C for 10 min and subsequently cooled on ice. FKB or potential interferents were prepared in MST buffer using a 2-fold serial dilution across 16 concentrations (10 μL per dilution). Next, 10 μL of the aptamer solution was mixed with 10 μL of each diluted sample and incubated for 5 min. The samples were then loaded into Monolith NT.115 standard capillaries. All experiments were performed independently in triplicate, and the data were analyzed using MO Affinity Analysis Software (Version 2.2.4).

### 2.7. AuNPs-Based Colorimetric Assay for Detection of FKB

A volume of 30 µL of 1 µM aptamer FKB-S was added to 100 µL of citrate-stabilized gold nanoparticles and thoroughly mixed. The mixture was incubated in a thermostatic metal bath at 37 °C for 14 min to facilitate aptamer conjugation. Following incubation, 30 µL of the FKB standard solution was added, and the mixture was incubated at room temperature for 5 min to allow target binding. Subsequently, 20 µL of a 400 mM NaCl solution was introduced to induce aggregation, which was followed by thorough mixing and a 5-minute incubation at room temperature. During this process, changes in the solution’s color were monitored, and the absorbance spectrum between 450 and 700 nm was recorded with specific attention to absorbance at 520 nm and 650 nm. A standard curve was constructed by plotting the FKB concentration on the x-axis and the absorbance ratio (A650/A520) on the y-axis, resulting in a linear equation, which was used to quantify FKB concentration.

## 3. Results

### 3.1. Generation of Aptamer for FKB

We employed a capture-SELEX strategy by immobilizing a 73 nt library containing a 30-nucleotide randomized (N30) region (DNA library) through hybridization with a 15-nt biotin-labeled complementary DNA (cDNA-bio) (Figure 1). The N30 region was fully randomized, and the two flanking constant primer binding regions can fold the library into an overall stem-loop structure. Upon binding to FKB, the library strand undergoes self-folding, detaching from the surface for collection and PCR amplification, enabling its use in subsequent selection rounds. This process was repeated until a pool enriched with high-affinity aptamers was obtained. A total of 27 rounds of selection were conducted, during which the concentration of FKB was progressively reduced from 500 μM to 5 μM. Additionally, various counter-targets, including structurally similar interferents and natural products, were incorporated to enhance the rigor of the selection process, thereby facilitating the isolation of aptamers with superior affinity and specificity. The concentration of target/counter eluted DNA was monitored throughout the selection using gel electrophoresis, allowing for the assessment of the selection’s progress (see Appendix A for detailed selection conditions).

In the first five rounds, as the FKB concentration decreased from 500 mM to 125 µM, the eluted DNA percentage by FKB significantly increased from 0.5% to 2.3% (Figure 2a). A gel elution assay was conducted to evaluate the target-binding affinity of the sub-library for FKB. The resulting dissociation constant (K_D_) value of 86.7 µM confirmed that FKB-binding aptamers had been successfully enriched through positive selection (Appendix A). Counter-SELEX was then performed prior to positive selection in order to remove aptamers binding to structurally similar interferents (e.g., FKB, FKA, isoliquiritigenin) that have the same functional groups or partial structural features as FKB [29]. Despite the increased counter-selection stringency, the fraction of counter-target eluted DNA was successfully reduced from 35% to 2%, while the FKB-eluted fraction increased from 5% to 10%. After the 19th round of selection, the K_D_ for FKB improved significantly, decreasing to 5.41 µM (Appendix A). However, the 19th pool demonstrated binding affinity not only for FKB but also for FKA and FKC with no detectable affinity for interferents such as myricetin, quercetin, aloe emodin, and isoliquiritigenin (Appendix A). These findings indicate a continued high prevalence of cross-reactive aptamers in the pool, particularly those targeting FKA and FKC, despite efforts to improve specificity. We speculate that this may be due to the excessive number of counter-targets, which likely impeded the effective binding of FKA and FKC to the DNA. To address this, a pure mixture of FKA and FKC was utilized in subsequent counter-SELEX. During rounds 20 to 27, the DNA eluted by FKA and FKC gradually decreased, while the DNA eluted by FKB steadily increased. Encouragingly, by the 27th round (Appendix A), the cross-reactivity of the library toward FKA and FKC was significantly reduced, prompting us to conclude the selection process.

### 3.2. High-Throughput Sequencing Analysis

High-throughput sequencing (HTS), used to analyze the abundance and enrichment fold of sequences, represents a practical and effective strategy for aptamer discovery [30,31]. Consequently, we conducted HTS on the Round 19 and Round 27 pools to identify potential aptamer candidates. Using the HTS results of the 19th pools as a benchmark, a total of 3306 families were identified within the 27th pools. Families with an abundance exceeding 0.1% were further selected for the calculation of their enrichment fold (Figure 2b). Within the 27th pools, the cluster FKB-1c emerged as the dominant family, comprising 6.3% of the sequences; however, its enrichment ratio was relatively low, reaching only 18.6-fold. Conversely, despite being smaller in size compared to FKB-1c, the clusters FKB-2c and FKB-3c were significantly enriched during the selection process, suggesting their potential for strong binding performance. Based on these observations, representative sequences from clusters FKB-1c, FKB-2c, and FKB-3c (named FKB-1, FKB-2, and FKB-3, respectively) were chosen for further affinity and specificity analysis.

We primarily employed the gel elution assay to assess the binding affinity of the selected candidates. The gel elution assay is a cost-effective and widely used strategy in Capture-SELEX for characterizing the affinity and specificity of aptamers. By utilizing gel electrophoresis to analyze the elution capacity of target molecules at gradient concentrations from the library, this method enables researchers to quickly assess the overall affinity and specificity of the library or individual candidate sequences. The results indicate that FKB-1 exhibited the strongest affinity with a K_D_ value of 2.87 μM. Following closely was FKB-2, which had a K_D_ of 9.97 μM. In contrast, FKB-3 demonstrated the weakest affinity with a K_D_ of only 40.5 μM (Figure 2c–e). We also employed the same gel elution assay to evaluate the specificity of these candidates (Figure 2f–h). No significant elution was observed for FKB-1 with 100 µM concentrations of myricetin, quercetin, aloe emodin, isoliquiritigenin, or even FKC and FKA. However, FKB-2 and FKB-3 exhibited some degree of cross-reactivity with FKC and FKA. Based on these findings, FKB-1 was selected as the optimal candidate for further characterization.

### 3.3. Characterization and Truncation of the Aptamer

The secondary structure of FKB-1, predicted using NUPACK, is depicted in Figure 3a. It adopts a hairpin-loop configuration, comprising one loop region (L1: nucleotides #30-47), two paired regions (P1: nucleotides #16-20 and #57-61, P2: nucleotides #22-29 and #48-55), and two junction sites (J1: nucleotide #21, J2: nucleotide #56). To verify aptamer binding, we first examined the K_D_ of the fluorophore (FAM) modified at the 5′ end of FKB-1 (FKB-1-5′FAM) using the microscale thermophoresis technique (MST) [32]. The K_D_ of FKB-1-5′FAM was determined to be 1.45 μM through MST analysis, which is slightly lower than the 2.87 μM value obtained using the gel-elution assay (Figure 3b). This difference may be attributed to the competitive displacement mechanism of the gel-elution assay, where the aptamer’s binding to the target competes with cDNA, typically resulting in higher K_D_ values. The G-rich sequence of FKB-1, with guanine accounting for 40% of the random region, suggests the potential formation of intramolecular G-quadruplexes. Analysis of the aptamer sequence using the QGRS Mapper yielded a G-score of 13, supporting the likelihood of G-quadruplex structure formation. Therefore, circular dichroism (CD) spectrometry was carried out to investigate the formation of G-quadruplexes. The CD spectrum of the aptamer exhibited a negative peak at 250 nm and a positive peak at 280 nm (Figure 3c), which is characteristic of a parallel G-quadruplex structure [33,34]. Upon the addition of FKB, the intensity of the positive peak increased, suggesting enhanced structural stability. G-quadruplex structures are a common characteristic of aptamers that bind to their targets. Based on this, we hypothesized that the binding domain of FKB-1 is located within the hairpin loop, which forms the G-quadruplex. Consequently, we further truncated the FKB-1 sequence, preserving its hairpin stem-loop structure (highlighted within the blue box in Figure 3a), and generated a simplified aptamer named FKB-S.

The three-dimensional structure of FKB-S, predicted using AlphaFold3 (Figure 3d), reveals that it retains a hairpin configuration similar to that of FKB-1. Notably, the bases within the hairpin loop form hydrogen bonds and stacking interactions, resulting in a compact structure with a potentially functional binding pocket. To further evaluate its binding performance, we characterized the affinity of FKB-S for FKB and its specificity against interferents using MST. The results demonstrated that FKB-S effectively retained its affinity for FKB with a K_D_ of 280 nM (Figure 3e). This represents a 5.2-fold improvement in binding affinity compared to FKB-1, which was likely due to reduced interference from the primer sequences. MST measurements further showed that FKB-S had low affinity for FKA and FKC and no detectable binding to other interfering substances (Figure 3f). To investigate the potential binding mechanism, molecular docking studies were conducted using HDOCK. Ten docking poses were generated, and the most favorable docking conformation, along with the corresponding interactions, is illustrated in Figure 3g. The results revealed that FKB interacts closely with FKB-S through six key bases (G18, G20, A22, T38, G42, and G43), forming four hydrogen bonds and four hydrophobic interactions via π–π stacking. These findings highlight that FKB-S, an aptamer with strong affinity and exceptional specificity, was successfully identified from unmodified DNA libraries. 

### 3.4. Principle of the Aptamer/AuNP Aggregation-Based Colorimetric Assay

To assess the application potential of the aptamer FKB-S in detecting FKB, we developed a colorimetric detection method leveraging the aggregation-induced color change in gold nanoparticles (AuNPs). Figure 1 illustrates the working principle of this sensor. Colloidal AuNPs are stabilized by adsorbed citrate ions, which confer a negative surface charge, preventing aggregation caused by van der Waals forces and maintaining the solution’s characteristic red color. Under high-salt conditions, this stabilization is disrupted, leading to AuNP aggregation and a color shift to purple. When the aptamer, which is negatively charged, binds to the positively charged surface of AuNPs, it increases the negative charge on the surface, thereby stabilizing the particles and preserving the red color even in high-salt conditions. However, upon the introduction of FKB, the aptamer binds specifically to FKB, reducing its capacity to protect the AuNPs from salt-induced aggregation. Consequently, the solution color transitions from red to purple with the absorbance change correlating to the concentration of FKB.

### 3.5. Optimization of the Detection Conditions

Citrate-stabilized gold nanoparticles (AuNPs) were synthesized through the reduction of HAuCl_4_ by sodium citrate in an aqueous solution. These 15 nm diameter AuNPs, which appear red in the absence of any additional solvent, exhibit a peak absorbance at 520 nm. Upon incubation with 400 mM NaCl, the negatively charged chloride ions (Cl^−^) interact with and disrupt the surface charge of the AuNPs, resulting in their aggregation and a shift to a purple color due to an increase in absorbance at 650 nm (Figure 4a). Transmission electron microscope (TEM) images revealed that the aggregates formed in the presence of NaCl have a significantly higher density compared to the original AuNPs (Figure 4b,c).

To achieve a highly sensitive aptasensor, various parameters were examined, including the concentrations of NaCl and aptamers, as well as the reaction durations for NaCl-AuNPs as well as aptamer-AuNPs. Since the NaCl concentration directly influenced the aggregation of AuNPs, optimizing both the NaCl concentration and incubation time was crucial. Initially, AuNPs were mixed with equal volumes of NaCl solutions at different concentrations (150, 200, 250, 300, 350, 400 mM). After thorough mixing, their absorption spectra were recorded, as depicted in Appendix A. As the NaCl concentration increased, the absorption peak of AuNPs at 520 nm gradually decreased, while the absorption peak at 650 nm increased progressively. The ratio A650/A520 stabilized at 400 mM NaCl, making it the optimal NaCl concentration. Subsequently, the optimal reaction time for NaCl-AuNPs was determined. As illustrated in Appendix A, as the reaction time between AuNPs and NaCl increased, the absorption peak of AuNPs at 520 nm decreased, indicating AuNP aggregation. After 5 min, the A650/A520 ratio stabilized, establishing 5 min as the optimal reaction time.

The aptamer can strongly adsorb onto the surface of AuNPs, enhancing their stability against salt-induced aggregation. Therefore, the concentration of the aptamer is a critical factor influencing its protective effect. To determine the optimal concentration, varying amounts of FKB-S (0.2–1.2 μM) were mixed with AuNPs, which was followed by the addition of 400 mM NaCl. The Uv/Vis spectra of the resulting solutions are presented in Figure 4d,e. As the FKB-S concentration increased, the absorbance of AuNPs at 520 nm also increased, indicating enhanced protection against aggregation in the salt solution. When the FKB-S concentration reached 1 μM, the A650/A520 ratio stabilized. Thus, 1 μM was selected as the optimal concentration for the aptasensor system. Additionally, the effect of reaction time on the binding between FKB-S and AuNPs was evaluated. As shown in Figure 4f, with increasing reaction time, the absorbance peak of AuNPs at 520 nm rose, signifying an improvement in the aptamer’s ability to prevent AuNP aggregation. At 14 min, the binding reached equilibrium, and the absorbance value stabilized. Therefore, a reaction time of 14 min was chosen for the detection system. Furthermore, the incubation time between the aptamer and its target has been extensively studied in similar research with typical durations ranging from 5 to 10 min. This aligns with the 5-minute incubation time used in the MST-based affinity characterization conducted in this study (Figure 3e). Accordingly, a 5-minute incubation was adopted for FKB in subsequent experiments.

### 3.6. Evaluating the Analytical Performance of Aptasensor

Under optimized experimental conditions, we evaluated the linear detection range and limit of detection of the aptasensor using varying concentrations of FKB. Figure 5a illustrates the changes in the UV/Vis absorption spectra observed when FKB, at concentrations ranging from 0 to 5 μM, was introduced into the Aptamer/AuNPs system. With increasing FKB concentration, the aptamer’s protective effect on the AuNPs was progressively lost, resulting in nanoparticle aggregation and a corresponding increase in the A650/A520 ratio (Figure 5b). The corresponding linear relationship can be expressed as follows: y = 0.0817C + 0.3941 within the concentration range of 0.4 to 2 μM. Utilizing the 3σ/k criterion, where σ represents the standard deviation of ten blank tests and k is the slope of the linear plot, the detection limit (LOD) was assessed to be as low as 150 nM (equivalent to 43.46 ng/mL) (Figure 5c). The detection limit of this method is significantly lower than the reported hepatotoxicity threshold of FKB (LD_50_ = 32 μM), highlighting the exceptional sensitivity of the sensor [9]. The sensor’s specificity for distinguishing the target from non-target interferents, including FKA, FKC, myricetin, and quercetin, was rigorously tested (Figure 5d). The results indicated that the aptasensor exhibited negligible response to these interferents compared to the buffer or even lower signals toward the derivatives FKA and FKC, thereby demonstrating its high specificity (Figure 5e). Furthermore, the sensor possesses a favorable detection window. Upon the introduction of the analyte, the absorbance of the test solution promptly increased within 5 min and maintained stability for over 10 min.

## 4. Conclusions

In conclusion, this study presents a novel and efficient method for the detection of FKB, which is a hepatotoxic chalcone compound found in kava products. By isolating and optimizing a highly specific DNA aptamer (FKB-S) with strong binding affinity, we developed an aptamer-based AuNP colorimetric assay that offers the rapid, sensitive, and selective detection of FKB. The assay achieves a detection limit as low as 150 nM (equivalent to 43.46 ng/mL) and demonstrates high selectivity against structurally similar compounds, including other chalcones and flavonoids. Notably, this aptasensor provides results within as little as 10 min, significantly enhancing its practicality for time-sensitive applications. Its simplicity, requiring minimal preparation and no specialized equipment, combined with its rapid response time and cost-effectiveness, makes it ideal for the on-site safety monitoring of kava products, addressing the pressing need for efficient hepatotoxicity risk mitigation. Future developments could focus on integrating this assay into portable devices or multiplexed platforms, further enhancing its utility in field diagnostics and real-time monitoring scenarios.

## Figures and Tables

**Figure 1 sensors-25-00569-f001:**
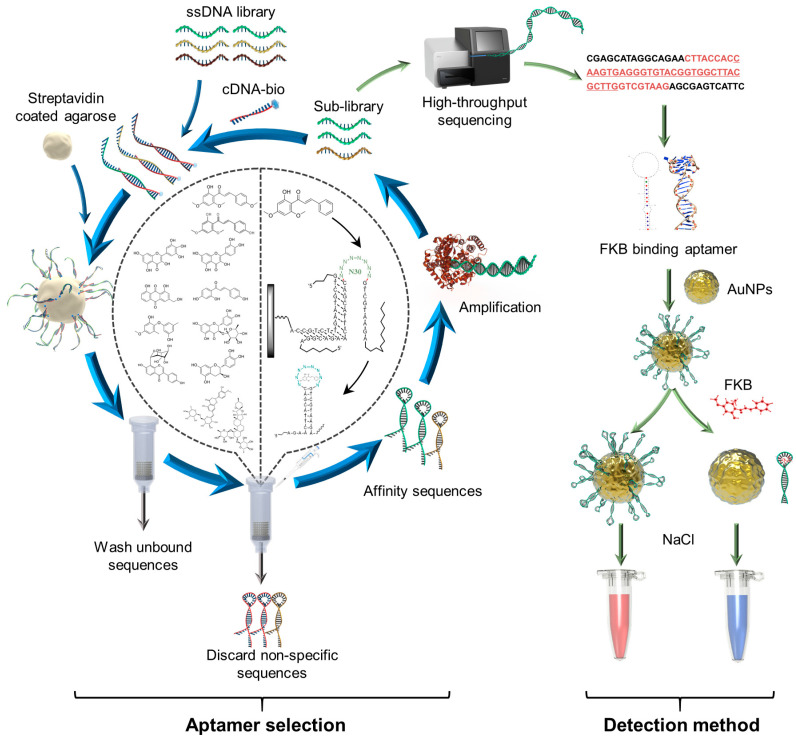
Schematic illustration of aptamer/AuNPs aggregation-based colorimetric assay.

**Figure 2 sensors-25-00569-f002:**
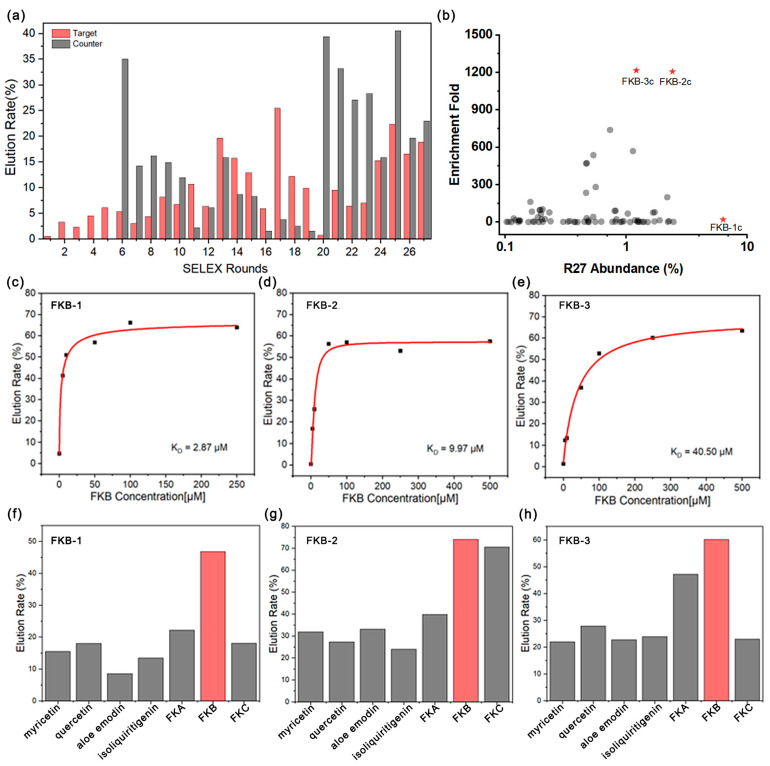
Monitoring the selection process. (**a**) The proportion of DNA eluted from the target and counter-targets in each round. (**b**) Enrichment folds of families with an abundance greater than 0.1% in round 27 compared to round 19. The families chosen are marked with red stars. Gel-elution assay characterization of the affinity for FKB-1 (**c**), FKB-2 (**d**), and FKB-3 (**e**). Gel-elution assay characterization of the specificity for FKB-1 (**f**), FKB-2 (**g**), and FKB-3 (**h**).

**Figure 3 sensors-25-00569-f003:**
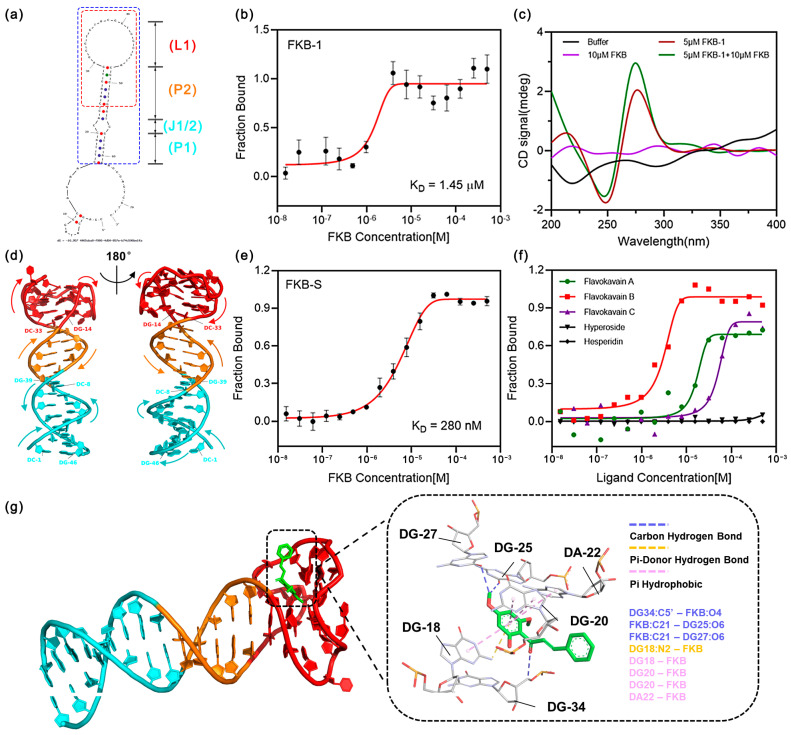
Characterization of aptamer. (**a**) Predicted secondary structure of FKB-1. The red box highlights the random sequence region, while the blue box indicates FKB-S. (**b**) FKB binding affinity of FKB-1 evaluated by MST. (**c**) CD spectra of buffer, free aptamer, FKB, and the aptamer in the presence of FKB. (**d**) Predicted 3D structure of the FKB-S aptamer. Arrows indicate the 5′-to-3′ chain direction. Residues from P1, P2, L1, and junctions (J1 and J2) are colored blue, orange, and red, respectively. (**e**,**f**) FKB binding affinity and specificity of FKB-S evaluated by MST. (**g**) Predicted binding pose of FKB with FKB-S.

**Figure 4 sensors-25-00569-f004:**
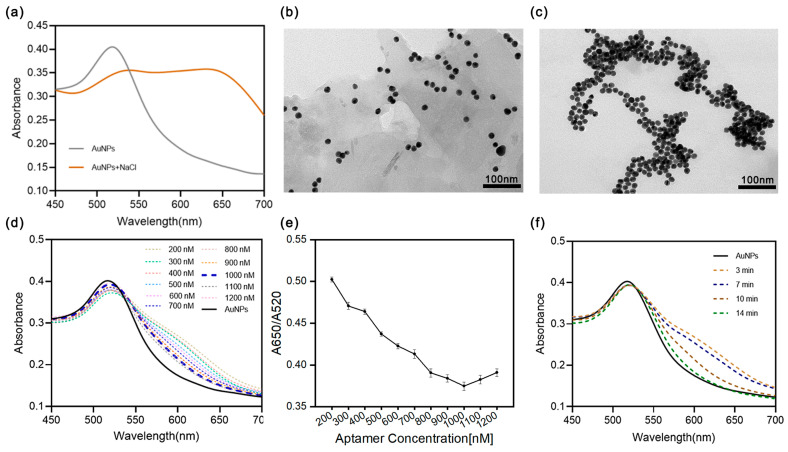
(**a**) UV/Vis spectra of AuNPs and aggregated AuNPs induced by NaCl. TEM image of (**b**) AuNPs and (**c**) aggregated AuNPs. (**d**) UV/Vis spectra of AuNPs at varying FKB-S concentrations with (**e**) the corresponding absorbance ratio (A650/A520). (**f**) Time-dependent UV/Vis spectral changes of AuNPs during incubation with 1 μM FKB-S.

**Figure 5 sensors-25-00569-f005:**
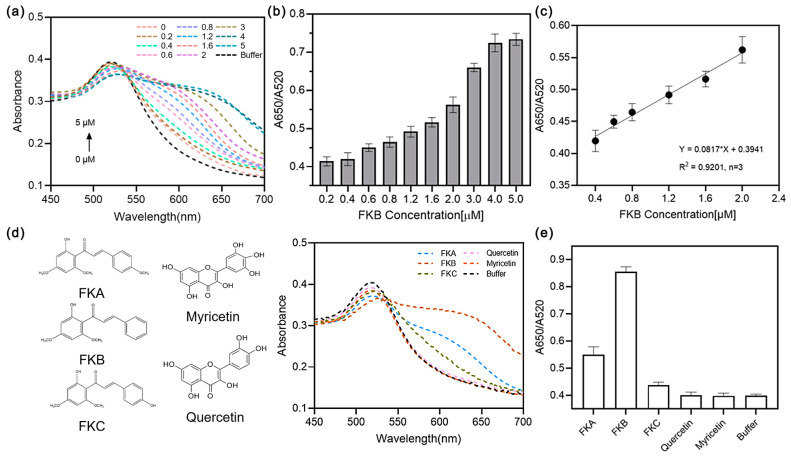
(**a**) UV/Vis spectra of aptasensor at different FKB concentrations (0, 0.2, 0.4, 0.6, 0.8, 1.2, 1.6, 2, 3, 4, 5 μM). (**b**) Corresponding absorbance ratios (A650/A520). (**c**) Calibration curve of A650/A520 versus FKB concentration. (**d**) Selectivity of the aptasensor in the presence of potential interferences with (**e**) the corresponding absorbance ratios (A650/A520). The error bars indicate the standard deviation for three detections.

## Data Availability

The data presented in this study are available on request.

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
