# Peer review of "A Label-Free Colorimetric Aptasensor for Flavokavain B Detection"

_sensors, 2025, doi:10.3390/s25020569_

Round 1

Reviewer 1 Report

Comments and Suggestions for Authors

In this work, DNA aptamers for flavokavain B have been selected, and a rapid and label-free colorimetric aptasensor for flavokavain B detection has been developed based on aptamer-induced gold nanoparticle aggregation. This research is innovative. The article can be accepted after revision.

1. Why chose the absorbance ratio (A650/A520) as the y-axis to develop the standard curve? Why not use the absorbance at A650 or A520 ?

2. The ruler is missing in Figure 4 b-c.

3. In Figure S2, it seems that the ratio of A650/A520 was increased with the NaCl concentration from 150 to 400 mM. Why not study the higher NaCl concentration than 400 mM? Is it reasonable that of 400 mM as the optimal NaCl concentration?

4. The error bar is missing in some figures, such as Figure 4e, Figure S2 (b and d).

5. In Figure 4e, how to get the conclusion that the binding reached equilibrium at 14 minutes. Because there lacks experiments with longer reaction time than 14 minutes for NaCl-AuNPs.

6. Line 373-376, where is the relevant experimental results or figures in this part?

7. In section 3.6, Figure 4 g-i is incorrect.

Reviewer 2 Report

Comments and Suggestions for Authors

In this manuscript, authors claimed to developed an aptamer targeting FKB with high binding affinity and turned it into a colorimetric assay. While the research work seemed solid, there are multiple improvements that authors need to make before the manuscript can be accepted. Comments as below:

1.      Authors mentioned the truncated aptamer has a dissociation constant of 228 nM from micro”a”cale thermophoresis (MST) (typo in abstract….), but through the entire manuscript I could not see the data showing the value… The closest one I can find was Figure 3e showing 280 nM. Authors need to explain..

2.      The KD values from Figure 2 c to e were from a different method and much higher than nanomolar range. Authors need to explain the difference and why was it important.

3.      What is the effective concentration of FKB in triggering hepatotoxicity? Authors need to compare the sensitivity of their aptamer to that level for evaluating the applicability.

4.      The CD data in Figure 3c showed that the aptamer maintained the hairpin structure before and after the binding with FKB. So the illustration of strand-displacement by FKB in Figure 1 was not accurate.

Comments on the Quality of English Language

Obvious typos in the manuscript. Serious editing is required.

Round 2

Reviewer 1 Report

Comments and Suggestions for Authors

The manuscript has been revised carefully. Therefore, I recommend its acceptance.

Reviewer 2 Report

Comments and Suggestions for Authors

Authors have addressed the question raised in the first review. I have no further question for the acceptance of this manuscript.